# Improved Quasi-Z-Source High Step-Up DC–DC Converter Based on Voltage-Doubler Topology

**DOI:** 10.3390/s22249893

**Published:** 2022-12-15

**Authors:** Toru Sai, Younghyun Moon, Yasuhiro Sugimoto

**Affiliations:** 1Faculty of Engineering, Tokyo Polytechnic University, Tokyo 243-0297, Kanagawa, Japan; 2Sugimoto Registered Consulting Engineer’s Office, Yokohama 224-0056, Kanagawa, Japan

**Keywords:** high step-up DC–DC converter, quasi-Z-source, voltage-doubler, high-side driver free, low voltage stress

## Abstract

The step-up DC–DC converter is widely used for applications such as IoT sensor nodes, energy harvesting, and photovoltaic (PV) systems. In this article, a new topological quasi-Z-source (QZ) high step-up DC–DC converter for the PV system is proposed. The topology of this converter is based on the voltage-doubler circuits. Compared with a conventional quasi-Z-source DC–DC converter, the proposed converter features low voltage ripple at the output, the use of a common ground switch, and low stress on circuit components. The new topology, named a low-side-drive quasi-Z-source boost converter (LQZC), consists of a flying capacitor (*C*_F_), the QZ network, two diodes, and a N-channel MOS switch. A 60 W laboratory prototype DC–DC converter achieved 94.9% power efficiency.

## 1. Introduction

Step-up DC–DC converters are widely used for applications that require high voltage converted from low input voltage, such as energy harvesting systems [1,2], IoT node operating systems on low-voltage battery cells [3], and photovoltaic (PV) systems for grid connection [4,5,6]. Although the necessary power and voltage for application is different, the same dc–dc converter topology is applicable and effective to these applications. The basic boost converter (BBC) is an established technique with a long history by which CR is determined by CR = 1/(1–D), where D is duty cycle. However, it has difficulty of control at high duty cycle when the duty ratio increases because of its nonlinear characteristics and narrow pulse width [7]. In addition, it suffers from the voltage stress on the semiconductor switches and conduction loss that is due to the long ON status of the switches during period D to hold the high CR status. Of course, the high CR can be simply obtained by the series connection of BBC, but it increases loss, cost, and volume; therefore, it is not smart option. To overcome these obstacles, alternative types of high step-up dc–dc converters have been proposed [8,9,10]. There are two approaches to achieve the high CR; one is the isolated type and the other is the non-isolated type. The isolated type obtained the high voltage by changing the secondary and primary turn ratios of the transformers. However, it produces inductor leakage and the necessity for the custom-made transformer, which additionally often requires measurements of unknown key electric and magnetic parameters. In addition, the bulky equipment is not suitable for the harvesting system and IoT node operation system. On the other hand, non-isolated converters are suitable for these applications because of both volume and cost. In this category, there are some promising converters introduced. R-J. Wai et al. [11], S-M. Chen et al. [12], and H-C. Liu et al. [13] proposed a coupled inductor type. However, it required the snubber circuits to suppress the leakage of inductors. In addition, the coupled inductors are not off-the-shelf components—the same as the transformer. Recently, a quasi-Z-source network has been installed in DC–DC converters [14,15,16,17,18,19,20,21,22,23,24]. Z-source networks typically consist of two inductors, two capacitors, and a diode that are connected to each other. Originally, Z-source networks were used for inverters to suppress shoot-through problems. L. Yang. et al. [14,15] applied the quasi-Z-source networks to the DC–DC converters. After that, several Z-source-type DC–DC converters were introduced. However, there are some drawbacks in previously introduced QZ DC–DC converters, such as low CR [14,15,16], an uncommon GND between input and output voltage [19,20,23], and complex implementation [18,21,22]. M. Veerachary et al. proposed the QZ boost converter [17], called a sixth-order quasi-Z-source DC–DC converter (SOQZCS). The achieved CR of the converter was (2–2D)/(1–2D). The operating duty cycle D is less than 0.5 to avoid narrow pulse-width control in the common GND configuration by simple implementation. However, it suffers from a large output ripple, requiring level shift circuits for the high-side switch, and high-voltage stress on circuit components. To eliminate these drawbacks, we reported a new QZ DC–DC converter [24].

Generally, a PV system has two approaches to connect the grid called DC-module type and AC-module type [25,26]. Since the output voltage of the one PV panel is small, that is from 15 V to 40 V [12], the DC-module type connects PV panels in series to build up the DC voltages for ensuring the grid voltage. However, the DC-module type has a drawback in principle for a partial shadow problem. When one PV panel is shadowed by obstacles such as clouds, leaves, and birds, the PV current becomes weak and the other series-connected PV-string currents are also weakened because they connect in series with each other. So, some avoiding methods are required [26]. On the other hand, the AC-module type connects the PV panels in parallel, and the partial shadow problem does not occur. However, instead, the AC-module type requires the high step-up DC–DC converter to ensure grid-connect voltage from the low output voltage of the one PV panel. The conventional AC-module installed BBC, so the aforementioned problems exist. A DC–DC converter for a PV system requires the following characteristics: the low input current for maximum point tracking (MPPT) control, low voltage stress at high step-up status, high efficiency, low cost, and reliability.

In general, to connect to the AC grid from the PV panels, an inverter circuit is inserted between the PV output and the grid to convert voltage from DC to AC. To the input terminal of the inverter, an electrolytic capacitor is connected to suppress the voltage ripple. The electrolytic capacitors are the most-aging components in the electric instruments, and they inflate the aging speed by their input voltage ripple. Of course, the electrolytic capacitor is a possible replacement of the film-type capacitor; however, the capacitance value of film-type capacitance is smaller and more expensive than the electrolytic capacitor. Therefore, the input voltage of the ripple is a significant parameter for the inverter from the viewpoint of cost and reliability. The proposed converter features high CR, low output voltage ripple, and low voltage stress for both semiconductor devices and flying capacitor, and is free from the high-side level shifter. A previously submitted paper [24] introduced the fundamental idea, the theoretical consideration, and the simulation results using a simulator. However, all elements used in the simulator are ideal elements. For example, the on resistance of transistor *M*, the parasitic resistance of inductor *L*, and the equivalent series resistance (ESR) of capacitor *C* are ignored. In addition, input voltage *V*_g_ and the output resistance *R*_o_ are ideal. So, the efficiency of the DC–DC converter was not evaluated. In this paper, we developed an actual 60 W laboratory prototype of the proposed DC–DC converter.

The operational principle is summarized in Section 2 and the topology comparison between [17] and LQZC are discussed in Section 3. Shown in the components selection, Section 4, are the measurement results of the conversion rate (CR), the measurement waveforms, and the measurement results of the efficiency. The measurement set-up and experimental results are shown in Section 4, and how the parasitic resistance of inductor *R*_DC_ affects the CR is considered theoretically and verified by a simulator in Section 5, and, finally, this work concludes in Section 6.

## 2. Circuits of the Proposed High Step-Up Converter

Figure 1a shows the proposed converter. The surrounded dot-line is a QZ network. It operates in two modes, Mode1 and Mode2.

Mode1: The equivalent circuit for this mode of operation is depicted in Figure 1b. The diode *D*_1_ and transistor *M* are turned ON and the diode *D*_2_ and *D*_3_ are turned OFF during this operation. The red circles denote the ON state of the switch transistor and diodes. Applying KVL to the closed loops, the following equations are obtained:*V*_g_ − *V*_L1_ + *V*_C2_ = 0(1)
*V*_g_ + *V*_C2_ − *V*_L2_ = 0(2)
*V*_CF_ = *V*_g_(3)

Mode2: The equivalent circuit for this mode of operation is depicted in Figure 1c. The *D*_2_ and *D*_3_ are turned ON and diode *D*_1_ and transistor *M* are turned OFF during this operation. Applying KVL to the closed loops, the following equations are obtained:*V*_L1_ = −*V*_C1_(4)
*V*_L2_ = −*V*_C2_(5)
*V*_g_ + *V*_C1_ + *V*_C2_ + *V*_CF_ = *V*_o_(6)

Applying the volt-second balance to inductor *L*_1_ using Equations (1) and (4), assume C_1_ = C_2_:(7)$$\left(V_{C2}+V_{g}\right)D-V_{C1}\left(1-D\right)=0$$
(8)$$V_{C1}=\frac{D\left(V_{C2}+V_{g}\right)}{1-D}$$
(9)$$V_{C1}=V_{C2}=\frac{DV_{g}}{1-2D}$$

By substituting Equation (9) for Equation (6), and using Equation (3), *V*_o_/*V*_g_ was obtained as follows:(10)$$\frac{V_{o}}{V_{g}}=\frac{2-2D}{1-2D}$$

Equation (10) shows the same CR (=*V*_o_/*V*_g_) as obtained by [17].

Figure 1d shows the key waveforms.

Figure 2 shows the CR of the BBC and the proposed converter. To eliminate narrow pulse-width control difficulty, the conventional converter [17] and the proposed converter [24] are designed to operate in a range from 0 to 0.5 of D, unlike BBT. The CR of Conv. [17] and Prop. [24] is obtained as 6.00 but that of BBT [7] becomes only 1.67 at D = 0.4.

## 3. Topology Comparison of Converters

In this section, the topology difference between the conventional converter and proposed converter is described. Figure 3 shows the three converters. Figure 3a shows the voltage-doubler [27]. Figure 3b,c are the Mode1 and Mode2 operations of it, respectively. Figure 3d shows a conventional converter [17]. Figure 3e,f are the Mode1 and Mode2 operations of it, respectively. Figure 3g shows the proposed converter [24]. Figure 3h,i are the Mode1 and Mode2 operations of it, respectively. The conventional converter topology can be regarded as the QZ replaced from the *S*_4_ in the voltage doubler. Considering other variations, we found the new topology, as shown Figure 3g. The proposed converter can be regarded as the QZ replaced from the *S*_3_ in the voltage doubler. By this modification, the proposed converter improved the output ripple, power loss, and voltage stress.

### 3.1. Output Voltage Ripple

The output voltage of the proposed converter is significantly reduced compared to the converter. The droop voltage of the output comes up when the output capacitor *C*_o_ disconnects the source voltage *V*_g_. This situation occurs in Mode2 in the conventional converter that is shown in Figure 3f. Since the output capacitor *C*_o_ connects to only the load current *I*_o_, the output voltage *V*_o_ droops at the rate determined by *V*_o_ = (*C*_o_/*I*_o_) t. As the high step-up DC–DC converter operates generally less than D = 0.5, the period of the operation time of Mode2 shown Figure 3f is longer than that of Mode1 shown Figure 3d. On the other hand, in the proposed converter, the period of time disconnected from source voltage *V*_g_ occurs in Mode1, as shown Figure 3h. This advantage results in that low EMI and the available small-output capacitor.

### 3.2. High Side Driver and Level Shifter

The switch *S*_3_ in Figure 3d is given by the N-channel MOS transistor [17], where the drain terminal of the N-channel MOS transistor is connected to *V*_g_ and the source terminal is connected to QZ. It means that this is necessary for the high-side driver and the level shift circuits for *S*_3_. The level shift circuits consume operating power, and some additional circuits are required for proper start-up of the DC–DC converter. In contrast, The LQZC is high-side driver-free because the source terminal of *S*_4_ is terminated to GND, as shown in Figure 3g. It can improve the efficiency and reduce the cost and complex design of the converter.

### 3.3. Voltage Stress

The proposed converter provides the reduction in voltage stress for the flying capacitor C_F_. The voltage stress of elements is shown in Table 1. In a conventional converter, the terminals of *C*_F_ are applied by *V*_o_ and *V*_g_ as shown in Figure 3e. Using Equation (10), the terminal voltage across *C*_F_ becomes:(11)$$V_{o}-V_{g}=\frac{2-2D}{1-2D}V_{g}-V_{g}=\frac{1}{1-2D}V_{g}$$

In contrast, the voltage stress of the *C*_F_ applied only *V*_g_. The voltage across *C*_F_ is determined as follows:(12)$$V_{o}-\left(V_{g}+V_{QZ}\right)=V_{o}-\left(V_{g}+V_{C1}+V_{C2}\right)\;=V_{o}-\left(V_{g}+\frac{2D}{1-2D}V_{g}\right)=V_{o}-\frac{1}{1-2D}V_{g}=V_{g}$$

This is obvious from Figure 3h and Equation (3). Consequently, the proposed converter can mitigate the voltage stress of *C*_F_. The stress voltages of other elements, *D*_1_, *D*_2_, *M*, *C*_1_, and *C*_2_, are same as those of a conventional converter [17].

## 4. Measurement Setup and Results

### 4.1. Measurement Setup

Figure 4 shows a photograph of the prototype. The values of the components and the part numbers are listed in Table 2. Here all capacitors are film type.

Figure 5 shows the block diagram of the measurement set-up. The clock frequency *f*_CLK_ is set to 100 kHz for all measurements. The duty cycle D is changed from 0.05 to 0.35 by function generator Textronix AFG3120. The drive voltage amplitude for the transistor *M* is set to 10 V. The input voltage *V*_g_ is set to 20 V, 30 V, and 40 V. The output voltage *V*_o_ connected the electric current load Array 3710 A, of which the output resistance *R*_o_ is set from 175 Ω to 500 Ω. The electrolytic capacitor *C*_IN_, 470 uF (200 V), is connected between the *V*_g_ and GND as an input filter. The waveform of the output voltage *V*_o_ and the flying capacitor *C*_F_ are measured by a high-voltage differential probe, Textronix P5200A, on 50:1 attenuation, and the inductor current *I*_L1_ is measured by the current probe Textronix P6021A. The input voltage *V*_g_ and clock signal CLK are measured by the passive probe Textronix TPP0250 on 10:1 attenuation.

### 4.2. Measurement Results

Figure 6 shows the steady-state measurement waveforms of *I*_L1_, *V*_o_, *V*_CF_, and CLK when *D* = 0.3 and *R*_o_ = 500 Ω, and set to *V*_g_ = 20 V, 30 V, and 40 V, respectively.

Figure 7 shows the relationship between the measured output voltage *V*_o_ and duty cycle D drawn together with the calculated gain by the dotted lines. The duty cycle D changed from 0.05 to 0.35 for input voltage *V*_g_ = 20 V, 30 V, and 40 V, respectively, on *R*_o_ = 500 Ω. The measurement results agree well with the calculated results. When the duty cycle D is small, the output voltage *V*_o_ is slightly low because of the forward voltage *V*_F_ of *D*_1_.

Figure 8 depicts the relationship between the output power *P*_o_ and the efficiency *η* using the data of Figure 7. Figure 9 shows the efficiency versus the output power *P*_o_ at *V*_g_ = 40 V and D = 0.2 when *R*_o_ varies from 500 Ω to 175 Ω by 25 Ω steps. The peak efficiency 94.9 % is obtained. Table 3 summarizes the performance comparison of Conv. [17] and this work.

## 5. Discussion

In this section, the effect of the parasitic element of the proposed converter is discussed. Generally, the equivalent series resistance (ESR) of the capacitors is relatively smaller than the DC resistance (*R*_DC_) of the inductor [28,29,30]. So, we focus on the parasitic element of the inductors. Figure 10 shows the proposed converter, which includes the parasitic resistance depicted by the blue resistor symbol.

Considering the steady-state condition, the average current through a capacitor operating in a periodic steady state is zero and the average current through an inductor operating in a periodic steady state is zero [7]; therefore, the average inductor current *I*_L1_ can be written as follows.
(13)$$I_{L1}=I_{g}-I_{o}$$

By substituting Equation (10) with Equation (13) and using *V*_g_*I*_g_ = *V*_o_*I*_o_, the following equation is obtained.
(14)$$I_{L1}=\left(\frac{V_{o}}{V_{g}}-1\right)I_{o}=\frac{1}{1-2D}I_{o}$$

Equation (14) indicates that the inductor current *I*_L1_ is larger than the output current *I*_o_. For example, *I*_L1_ = 5*I*_o_ when D = 0.4 and *I*_L1_ = 1.67*I*_o_ when D = 0.2. This means that the equivalent series resistance (*R*_DC_) of inductors affects the efficiency of the converter and CR, too. Here, we focus on how much the CR is affected by the *R*_DC_ of *L*_1_ and *L*_2_.

In Mode1, the voltage expressions obtained using KVL are:(15)$$V_{g}-V_{L}-V_{r}+V_{C}=0$$
where *V*_r_ = *R*_DC_ × *I*_L_.

In Mode2, the voltage expressions obtained using KVL are:(16)$$-V_{L}-V_{r}-V_{C}=0$$

Applying the volt–second balance to inductor L:(17)$$(V_{g}+V_{C}-V_{r})D-\left(V_{C}+V_{r}\right)(1-D)=0$$
(18)$$V_{C}=\frac{DV_{g}-V_{r}}{1-2D}$$

Compared to Equation (9), Equation (18) indicates that when the voltage of capacitor *V*_c_ is reduced by *V*_r_, it results in reducing *V*_o_. This is attributed to the voltage across inductors *V*_L_ being reduced in Mode1 by *R*_DC_.

By substituting Equation (18) for Equation (6), *V*_CF_ = *V*_g_, *V*_o_/*V*_g_ is obtained as follows:(19)$$\frac{V_{o}}{V_{g}}=\frac{2-2D}{1-2D}-\frac{2V_{r}}{(1-2D)V_{g}}$$

To verify Equation (19), the output voltage *V*_o_ was checked by the PSIM [30] simulator when conditions changed. In the simulator circuit, the output resistance *R*_o_ is set to 100Ω, and *R*_DC_ are added to *L*_1_ and *L*_2_, respectively, in Figure 11. In this situation, four cases are tested. Figure 12 shows the simulation results of output voltage *V*_o_ in four cases. The blue line is output voltage *V*_o_, and the red line is input voltage *V*_g_. Table 4 shows the results of calculations and the simulation results.

Calculated numerical equations in Case4 are shown as follows by Equations (19) and (14); *I*_L1_ = *I*_o_/(1–D) = 1.67*I*_o_*, I*_o_
*=* 127 V/*R*_o_
*=* 127 V/100 Ω, and *V*_o_ becomes:(20)$$\begin{array}{l} \frac{V_{o}}{V_{g}}=\frac{2-2D}{1-2D}-\frac{2I_{L1}R_{DC}}{(1-2D)V_{g}}\\ =\frac{2-2\times 0.2}{1-2\times 0.2}-\frac{2\times 1.67\times \frac{127\;V}{100\;\Omega }\times 100\;m\Omega }{(1-2\times 0.2)48\;V}=2.652 \end{array}$$
(21)$$V_{o}=48\;V\times 2.652=127.3\;V$$

Table 4 shows that the calculation results correspond well to the simulation results.

## 6. Conclusions

This article has introduced a new high step-up DC–DC converter. The LQZC realizes a low output ripple, is free from the use of a level shifter and a high-side switch, and is plus the low stress on a flying capacitor *C*_F_. The proposed converter makes the following contributions: (i) by the low output ripple, a reliability and cost reduction for the DC–DC converter itself and the inverter circuit for the PV system because the low ripple voltage reduces the size of the capacitors and its aging. (ii) By omitting the level shifter circuit, the consuming loss is definitely reduced because the level shifter circuits and their accompanied circuits are not necessary. (iii) By reducing the voltage stress of the components, the equipment volume becomes small and reduces cost. The achieved efficiency of the converter was more than 94.9% in this prototype. Although the prototype design focuses on a PV application, the proposed architecture can be applied to applications that require the high step-up DC–DC converter, such as energy harvesting and low voltage battery systems for IoT sensor nodes, by power scaling down.

## Figures and Tables

**Figure 1 sensors-22-09893-f001:**
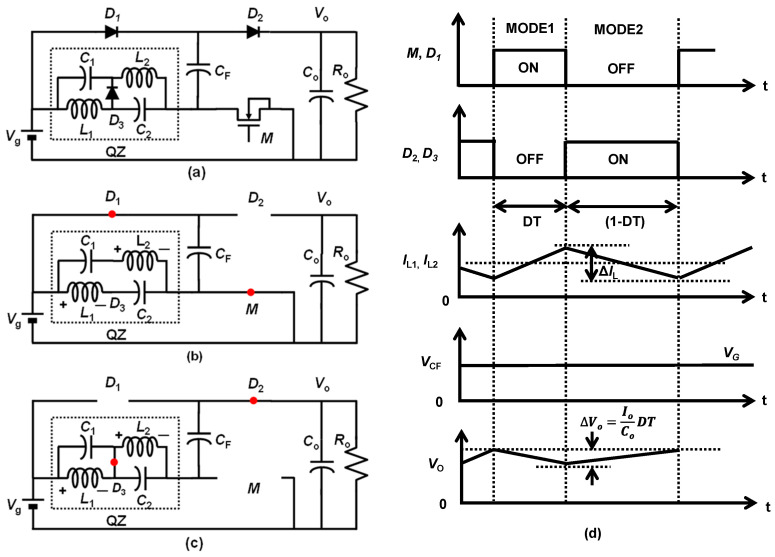
(**a**) Proposed high step-up DC–DC converter (LQZC). (**b**) Mode1 operation. (**c**) Mode2 operation [24]. (**d**) Key waveforms. Reprinted/adapted with permission from Ref. [24]. Copyright 2022, IEICE.

**Figure 2 sensors-22-09893-f002:**
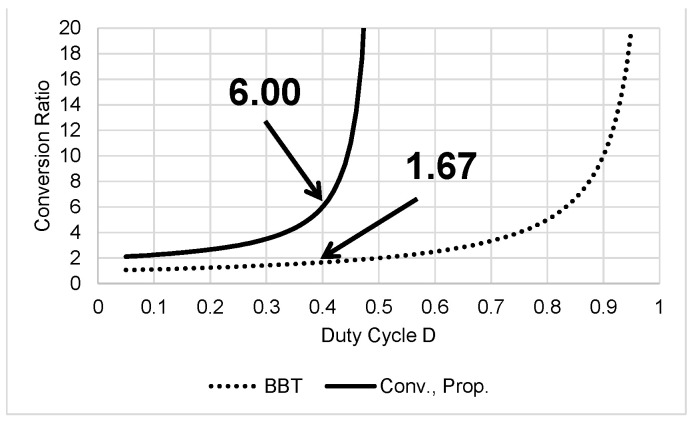
CR of BBC [7] and Conv. [17], Prop. [24] converters.

**Figure 3 sensors-22-09893-f003:**
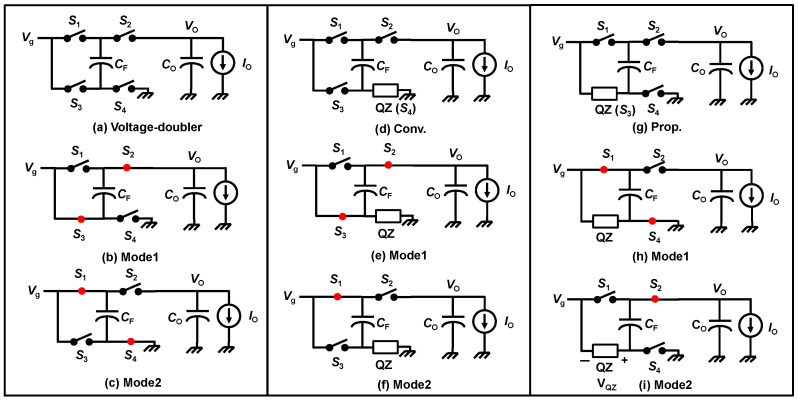
Comparison of converters: (**a**) voltage doubler, (**b**) Mode1 of voltage doubler, (**c**) Mode2 of voltage doubler, (**d**) Conv., (**e**) Mode1 of Conv., (**f**) Mode2 of Conv., (**g**) Prop., (**h**) Mode1 of Prop., and (**i**) Mode2 of the Prop. Reprinted/adapted with permission from Ref. [24]. Copyright 2022, IEICE.

**Figure 4 sensors-22-09893-f004:**
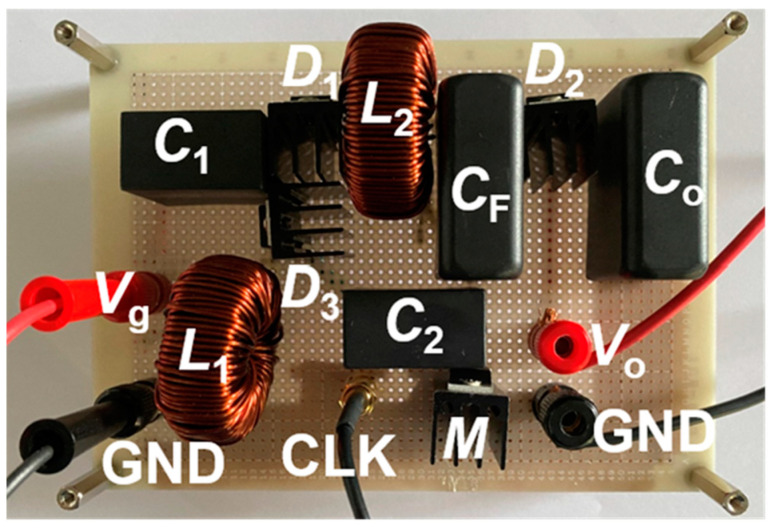
Photograph of prototype.

**Figure 5 sensors-22-09893-f005:**
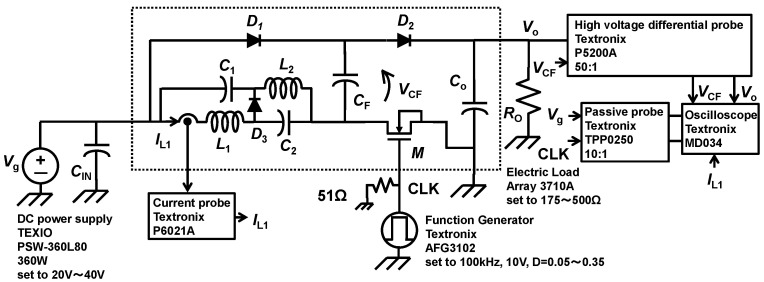
Measurement set-up.

**Figure 6 sensors-22-09893-f006:**
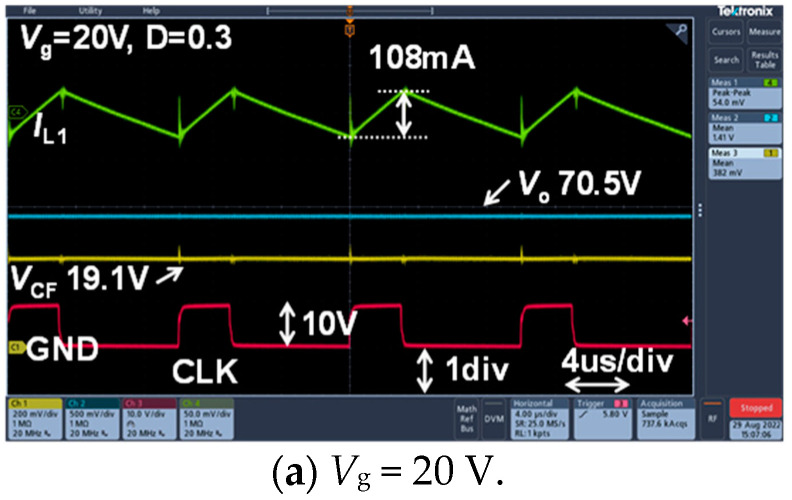
Experimental waveforms. Green line: *I*_L1_. Blue line: *V*_o_. Yellow line: *V*_CF_. Red line: CLK.

**Figure 7 sensors-22-09893-f007:**
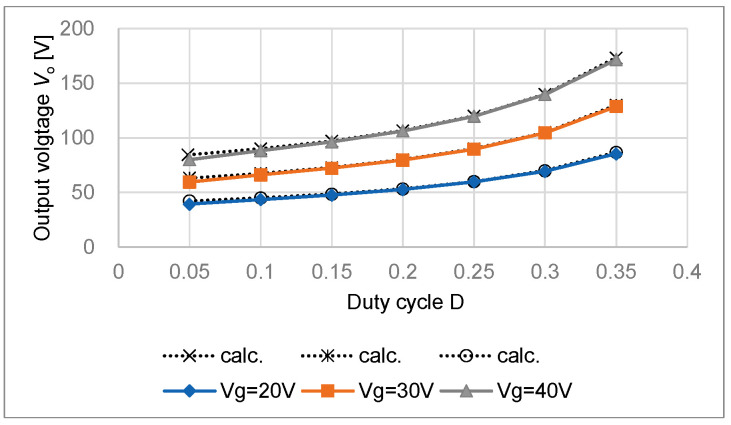
Output voltage *V*_o_ and duty cycle D.

**Figure 8 sensors-22-09893-f008:**
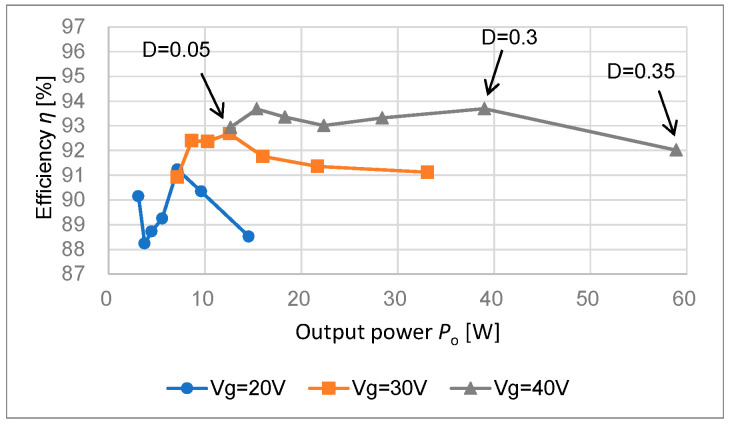
Efficiency η using data of Figure 7.

**Figure 9 sensors-22-09893-f009:**
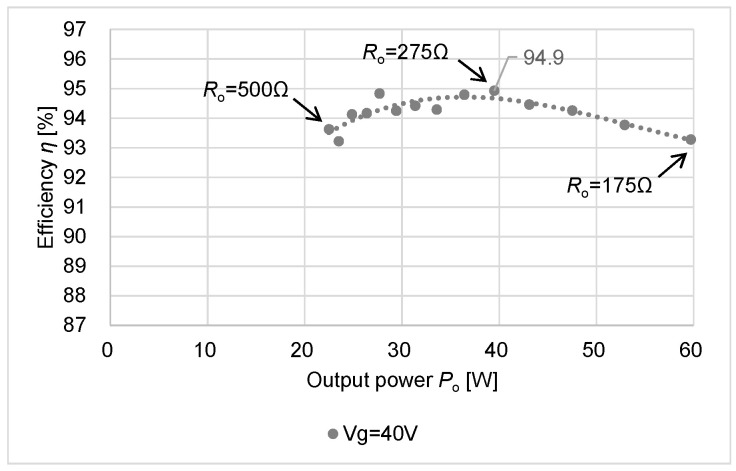
Peak efficiency η.

**Figure 10 sensors-22-09893-f010:**
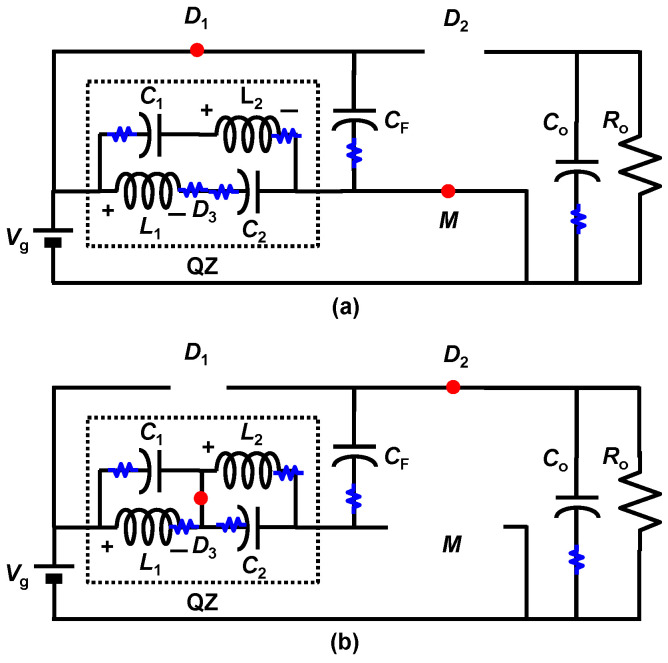
Proposed high step-up DC–DC converter (LQZC) with parasitic resistances. (**a**) Mode1. (**b**) Mode2.

**Figure 11 sensors-22-09893-f011:**
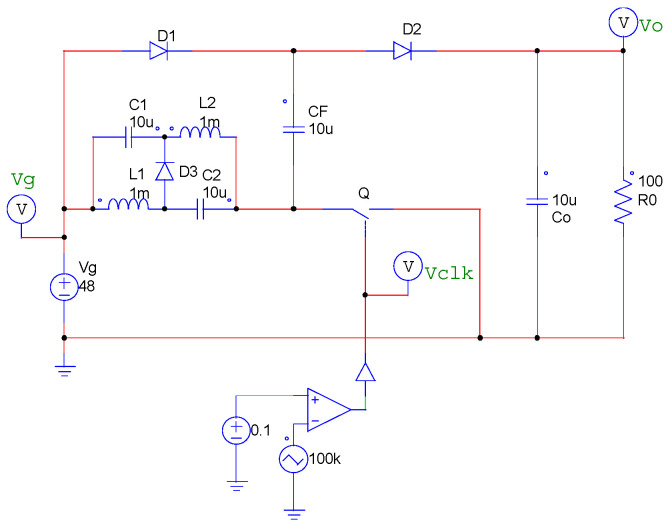
Simulation circuits.

**Figure 12 sensors-22-09893-f012:**
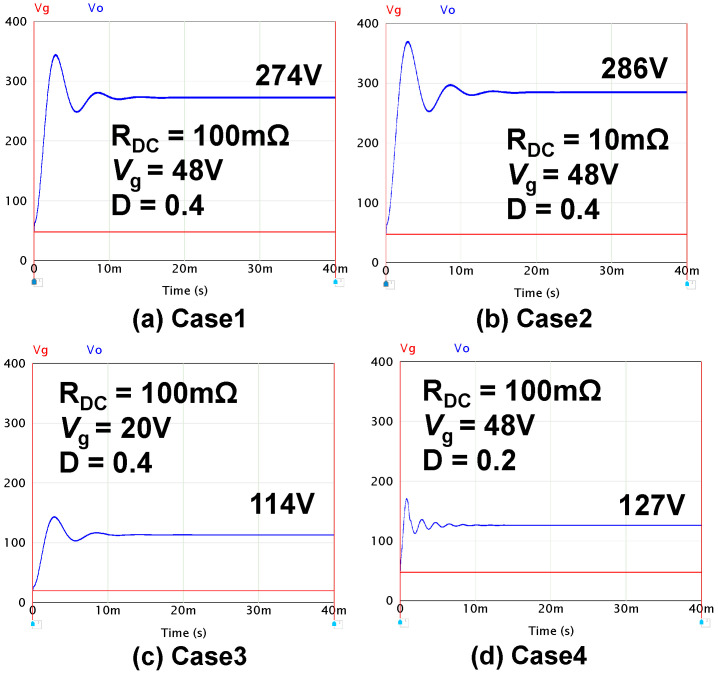
*V*_O_ and *V*_g_. (**a**) Case1. (**b**) Case2. (**c**) Case3. (**d**) Case4.

**Table 1 sensors-22-09893-t001:** Voltage stress of components.

Components	Conv.	Prop.
*D*_1_, *D*_2_	*V*_O_ − *V*_g_	*V*_O_ − *V*_g_
*M*	$\frac{1}{1-2D}V_{g}$	$\frac{1}{1-2D}V_{g}$
*C*_1_, *C*_2_	$\frac{D}{1-2D}V_{g}$	$\frac{D}{1-2D}V_{g}$
*C* _F_	$\frac{1}{1-2D}V_{g}$	*V* _g_

**Table 2 sensors-22-09893-t002:** Components specification.

Components	Values and Main Parameters	Parts No.
*L*_1_, *L*_2_	1 mH, 5 A, R_DC_ = 111 mΩ	TAMURA NAC-06-1001
*C*_1_, *C*_2_	10 uF, *I*_RMS_ = 7 A, ESR = 10 mΩ	VIHSAY MKP1848C61050JK2
*C*_F_, *C*_O_	20 uF, *I*_RMS_ = 8 A, ESR = 9 mΩ	VIHSAY MKP1848C62050JP2
*D*_1_, *D*_2_, *D*_3_	300 V, 3 A, *V*_F_ = 1.66 V	POWER INTEGRATIONS LQA30T300
*M*	650 V, *I*_D, CNT_ = 12 A_MAX_, 190 mΩ	INFINEON IPP65R190CFD7

**Table 3 sensors-22-09893-t003:** Performance summaries.

Items	M. Veerachary ICSETS 2019 [11]	This Work
CR	$\frac{2-2D}{1-2D}$	$\frac{2-2D}{1-2D}$
Output Voltage Ripple	$\frac{V_{O}}{C_{O}}\left(1-D\right)T$	$\frac{V_{O}}{C_{O}}DT$
Voltage Stress of C_F_	$\frac{1}{1-2D}V_{g}$	*V* _g_
High Side Driver and Level Shifter	Necessary	Unnecessary
Efficiency	N/A	94.9%

**Table 4 sensors-22-09893-t004:** Comparison of calculation and simulation results in four cases.

Case	*R* _DC_	*V* _g_	D	*V*o_calc.	*V*o_sim.
Case1	100 mΩ	48 V	0.4	274.3 V	274 V
Case2	10 mΩ	48 V	0.4	286.6 V	286 V
Case3	100 mΩ	20 V	0.4	114.3 V	114 V
Case4	100 mΩ	48 V	0.2	127.3 V	127 V

## Data Availability

Not applicable.

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
