# Peer review of "Improved Quasi-Z-Source High Step-Up DC–DC Converter Based on Voltage-Doubler Topology"

_sensors, 2022, doi:10.3390/s22249893_

Round 1

Reviewer 1 Report

Based on the voltage-double circuits, a new topological quasi-Z-source (QZ) high  step-up dc-dc converter for the PV system is proposed in the paper. Finally, a 60W laboratory prototype DC-DC converter can achieve 94.9% power efficiency. The contents of the paper do have some interests, but the reviewer also has some comments.

1.      The contribution of the paper is not clear enough, the authors should highlight why the topology of the QZ high set-up dc-dc converter for the PV system is proposed.

2.      The paper is not well written, and the contribution of the paper is not clear. It should be explained clearly.

Reviewer 2 Report

This paper proposed a Z-Source High Step-Up DC-DC converter. The converter is verified by theoretical analysis and simulation with experiment setup. To make the paper even better, I have some comments listed below:

1.      Is SENSORS MDPI a good choice for this type of publication? In my opinion ENERGIES MDPI or ELECTRONIC MDPI would be a better choice for this article.

2.      LINE 15 –There should be space between number and unit. Please make a correction in the article.

3.      Better state of the art is required in the Introduction – please provide a better comparison with different DC-DC converters not only [11].

4.      LINE 48 – wrong color of the text

5.      Fig. 1 and Fig 3. have been used in publication [12] – please add reference to the article.

6.      Key waveforms of the proposed converter are required

7.      Fig. 4 - required space before.

8.      You have used VIHSAY MKP1848C6xxxx capacitors – please explain what is the current value in the table 2?

9.      Table 2 – “D1, D2 ,D2” – it should be “D1, D2 ,D3”.

10.  Please add information about diode and remove “N/A“ description – forward voltage drop is required

11.  Did you use low-side driver for MOSFET or did you directly connect it with AFG? What is the current value for the MOSFET in Table 2? Is it drain pulse current or continuous?

12.  Figs. 7-9 - I suggest to use approximation instead of interpolation for the data chart.  

Round 2

Reviewer 1 Report

The authors have corrected all the comments, so the paper can be accepted as current version.

Reviewer 2 Report

Thanks for addressing my comments in the review. I have no further remarks.
p.s. please add space between all numbers and symbols in the manuscript